# Combining Historical Remote Sensing, Digital Soil Mapping and Hydrological Modelling to Produce Solutions for Infrastructure Damage in Cosmo City, South Africa

**George van Zijl [1,\*], Johan van Tol [2], Darren Bouwer [3], Simon Lorentz [4] and Pieter le Roux [3]**

1   Unit for Environmental Sciences and Management, North-West University, Potchefstroom 2520, South Africa
2   University of the Free State, Nelson Mandela Avenue, Bloemfontein 9301, South Africa; vantoljj@ufs.ac.za
3   Digital Soils Africa, 2 Alabaster street, Port Elizabeth 6001, South Africa; darren@dsafrica.co.za (D.B.); pieter@dsafrica.co.za (P.l.R.)
4   Centre for Water Resources Research, Univ. of Kwa-Zulu-Natal, Pietermaritzburg 3200, South Africa; slorentz@srk.co.za
\*   Correspondence: george.vanzijl@nwu.ac.za; Tel.: +27-18-299-2335

**Abstract:** Urbanization and hydrology have an interactive relationship, as urbanization changing the hydrology of a system and the hydrology commonly causing structural damage to the infrastructure. Hydrological modelling has been used to quantify the water causing structural impacts, and to provide solutions to the issues. However, in already-urbanized areas, creating a soil map to use as input in the modelling process is difficult, as observation positions are limited and visuals of the natural vegetation which indicate soil distribution are unnatural. This project used historical satellite images in combination with terrain parameters and digital soil mapping methods to produce an accurate (Kappa statistic = 0.81) hydropedology soil map for the Cosmo City suburb in Johannesburg, South Africa. The map was used as input into the HYDRUS 2D and SWAT hydrological models to quantify the water creating road damage at Kampala Crescent, a road within Cosmo City (using HYDRUS 2D), as well as the impact of urbanization on the hydrology of the area (using SWAT). HYDRUS 2D modelling showed that a subsurface drain installed at Kampala Crescent would need a carrying capacity of $0.3$ m$^3 \cdot$h$^{-1} \cdot$m$^{-1}$ to alleviate the road damage, while SWAT modelling shows that surface runoff in Cosmo City will commence with as little rainfall as 2 mm$\cdot$month$^{-1}$. This project showcases the value of multidisciplinary work. The remote sensing was invaluable to the mapping, which informed the hydrological modelling and subsequently provided answers to the engineers, who could then mitigate the hydrology-related issues within Cosmo City.

**Keywords:** urban soils; hydropedology; MNLR; machine learning; Johannesburg; remote sensing

## 1. Introduction

Urban sprawl is set to continually put larger developmental pressure on our natural eco-systems, as larger portions of the world's increasing population will live in cities. Currently, 54% of the world's population are city dwellers, and this is projected to increase to 66% by 2050 [1], leading to a greater need for infrastructure development in the areas surrounding cities.

The relationship between hydrology and infrastructure development is interactive. The expansion of urban areas can lead to drastic changes in hydrological processes. For example, the generation mechanisms, magnitude, and timing of streamflow could be altered by infrastructure [2–4]. On the other hand, these changes in hydrological processes can lead to infrastructural damages when they

exceed construction thresholds [5,6]. Lately, the focus in urban development has been on water-sensitive design strategies, which not only aim to protect infrastructure from water damage, but also limit the occurrence of flood events, while striving to maintain pre-development hydrological dynamics [7]. In order to implement water-sensitive design strategies, the hydrological processes before development should be identified, characterized, and quantified.

The study of water movement in soils, or hydropedology [8], utilizes soil properties to characterize the hydrological behavior within the soils. This is possible because water is a primary agent in soil genesis, and directly influences the formation of soil properties, which in turn carry unique signatures as to how they are formed [9–11]. Conversely, hydrological processes are significantly influenced by the spatial distribution of soil properties [12], as soil transmits, stores, and reacts with water [13].

To capture the spatial distribution of soil properties, hydropedological studies generally use soil maps [14,15]. However, in urban areas that are already developed, conventional soil maps are impossible to create due to the limited access to undisturbed soil observation positions. Therefore, older developments have double disadvantages, firstly by not being developed in a water-sensitive way, and secondly, when hydrology-related problems present themselves, the conventional methods to create a soil map for a hydropedological study are impossible to carry out. Digital soil mapping (DSM; [16]) provides the answer, as it uses environmental covariates (digital elevation models and satellite images) to predict the soil distribution at unobserved areas, based on the observed soils and their relation to the covariates.

This paper presents how historical remote sensing was used to provide the pre-urban land cover required to use DSM methods to create a soil map for Cosmo City, a suburb in Johannesburg, South Africa. By using Landsat satellite imagery from 2004, before the development commenced, a natural-conditions soil map could be created using DSM methodology. This map was then used as input to model the hydrology of the suburb, which allowed for the identification and prescription of mitigation measures at Kampala Crescent—a road where subsoil water caused serious road damage. Furthermore, the influence of the infrastructure development on the hydrology of the area was quantified by hydrological modelling of the entire suburb. This study differs from previous studies which used remote sensing data for hydrological models [17–19], as the Landsat images were used to map the soil, which were the input into the hydrological model, rather than using Landsat images for land cover or surface sealing indications.

## 2. Materials and Methods

### 2.1. Site Description

The study site consisted of Cosmo City, a suburb of Johannesburg (Figure 1), where construction started in 2005. The geology of the study site is granite and gneiss of the Halfway House Granite formation [20], wherein Leptosols, Stagnosols, and Acrisols are the main soil reference groups found [21]. The natural vegetation is the Egoli Granite Grassland [22], however, very little of the natural vegetation remains with the construction of Cosmo City. Topographically, the study site lies on the Highveld of Southern Africa, at heights above sea level between 1400 m and 1540 m [23]. The terrain is hilly, with slopes of up to 12%, but with the majority of hillslopes having an average slope of below 5%. Johannesburg's climate is typical of the subtropical highland, with hot days during summer (average maximum temperature = 25 °C) and cold nights during winter (average minimum temperature = 4 °C). The mean annual precipitation is 713 mm, mostly from thundershowers during the summer months.

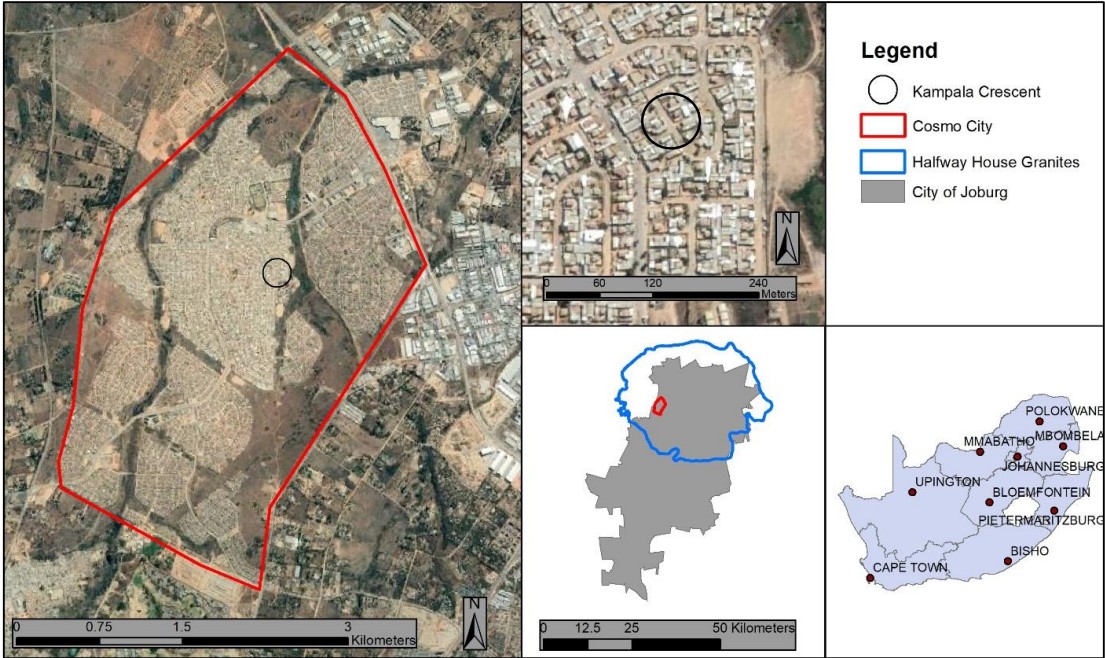

**Figure 1.** The location of Cosmo City and Kampala Crescent.

*2.2. Methodology*

2.2.1. General Survey

Soil observations were made across Cosmo City by making observations by soil auger, profile pits, and spot observations in the area. The observations were made on transects in areas where the soil was deemed to be undisturbed. Within Cosmo City, 61 soil observations were made to refusal or a depth of 3200 mm (Figure 2). Soils were described per horizon with soil texture, structure, color, redox morphology, stone content, and transitions being noted. Observations were classified to the soil family level according to the South African Soil Classification System [24], and to the reference group level of the World Reference Base [25]. Undisturbed samples for soil physical measurements were taken of selected soil horizons. Samples were analyzed for texture with the pipette method, bulk density by weighing a fixed volume sample and $K_{sat}$, drained upper limit and saturation point by the core lab infiltration method. At specific locations where undisturbed samples could not be taken (e.g., horizons with a large stone content), $K_{sat}$ was measured in situ with the Guelph permeameter.

2.2.2. Digital Soil Mapping

To create a soil map, environmental covariates were collected for the entire Halfway House Granite area, to use as ancillary variables in the mapping process. These layers included wet and dry season Landsat 5 images [26], taken on 10 April 2004 (wet season) and 31 July 2004 (dry season) respectively, before construction on Cosmo City began, and the 30 m SRTM DEM [26]. Covariate layers were resampled to have the same grid extent at a resolution of 30 m. Secondary covariate layers were derived from the Landsat 5 and DEM layers in SAGA-GIS [27], which included: slope, profile curvature, planform curvature, aspect, topographic wetness index, flow accumulation, altitude above channel network, relative slope position, and multi resolution index of valley bottom flatness. From the wet and dry season satellite images the normalized difference vegetation index (NDVI) was derived.

For modelling purposes the Cosmo City soil observation database was combined with two other soil observation databases within the Halfway House Granites (Figure 3) [28,29]. This allowed for the soil observation database to be enlarged with 212 observations (70 from Van Zijl and Bouwer 2012 [29] and 142 from Van Zijl et al. (2019)) [28]. Therefore, the final Halfway House Granites database

consisted of 273 soil observation points. Hydropedological soil forms [30] were derived from the soil form classifications according to Table 1.

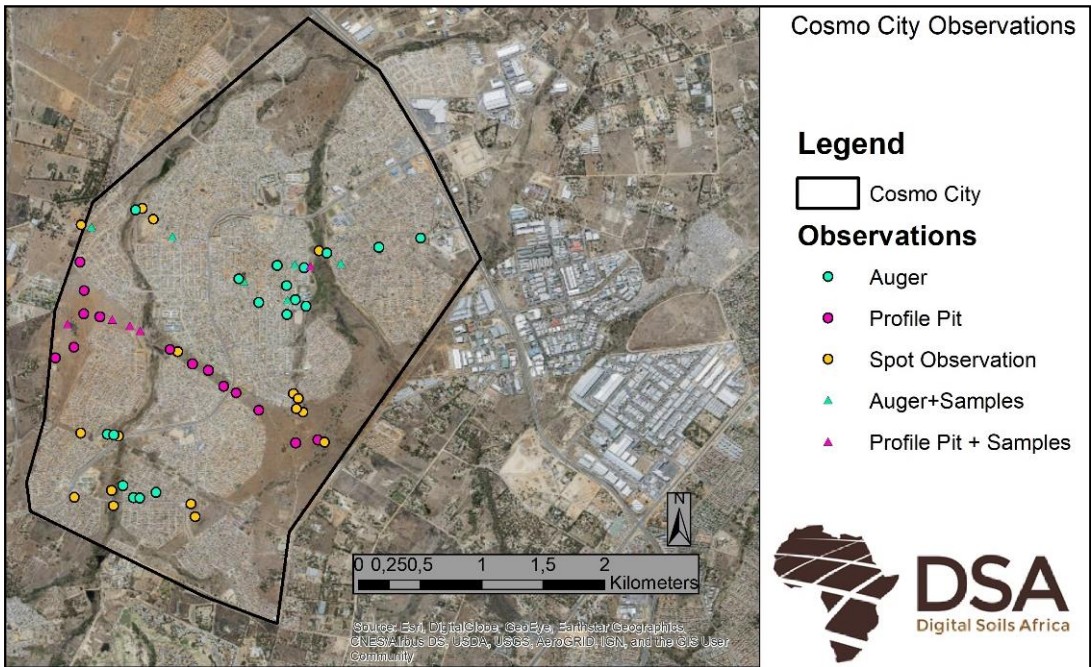

**Figure 2.** Soil observations for the Cosmo City soil observation database.

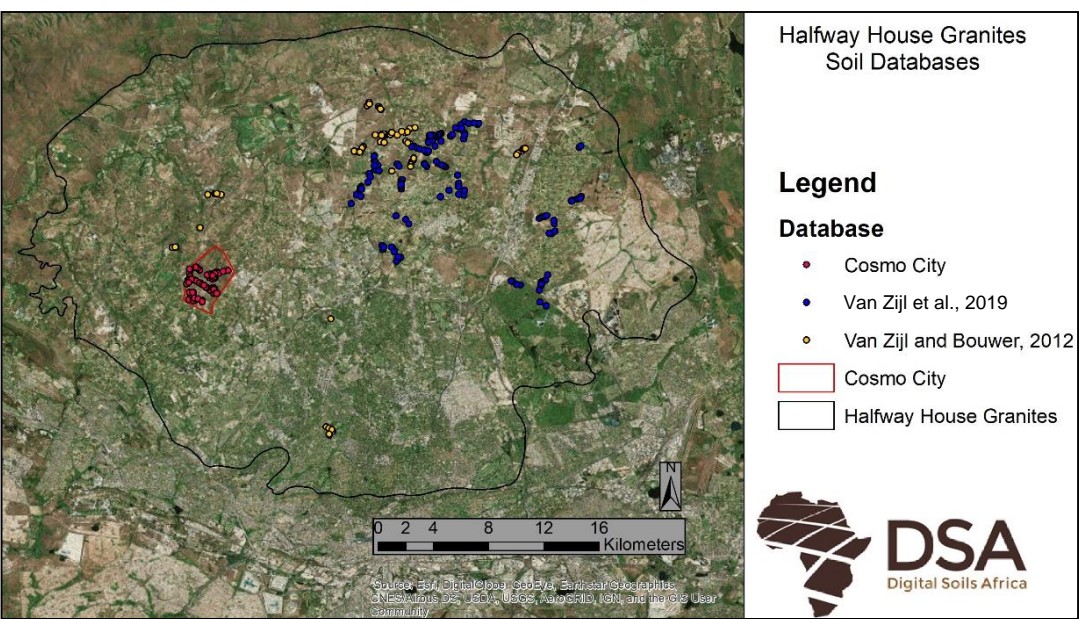

**Figure 3.** The soil observations of the three Halfway House Granites soil observation databases.

**Table 1.** Description of the hydropedological soil map units.

| Hydropedological Soil Form | Soil Forms (SCWG 1991) | Reference Groups (IUSS 2015) | Defining Characteristic |
|---|---|---|---|
| Deep Recharge | Clovelly, Constantia, Griffin, Hutton, Shortlands | Acrisols, Nitisols | Soil profiles showing no signs of wetness in the profile |
| Shallow Recharge | Mispah, Glenrosa, Mayo | Leptosols | Shallow soils with chromic colors in the topsoil |
| Interflow (A/B) | Constantia, Kroonstad, Longlands, Sterkspruit, Wasbank | Stagnosols, Planosols, Plinthosols | Signs of wetness between top and subsoil |
| Interflow (Soil/Bedrock) | Avalon, Bainsvlei, Bloemdal, Dresden, Fernwood, Glencoe, Pinedene, Tukulu, Westleigh | Acrisols, Stagnosols, Arenosols, Plinthosols, | Signs of wetness at soil bedrock interface |
| Saturated Responsive | Katspruit, Rensburg | Gleysols | Gleyed subsoil |
| Shallow Responsive | Mispah, Glenrosa | Leptosols | Shallow soil with bleached colors in the topsoil |

The combined soil database was divided into training and evaluation datasets by stratified random sampling using soil form as a stratifier, with 25% of the observation points of each hydropedological soil form being included into the evaluation dataset. Care was taken to include 25% of each hydropedological soil form of the Cosmo City database into the evaluation set. However, as no shallow responsive observations occur in this dataset, they were not included in the evaluation dataset for Cosmo City. The training dataset consisted of 203 observations, with 70 observations in the evaluation dataset. For the Cosmo City dataset, the training observations totaled 44, while the evaluation observations made up the remaining 17 observations.

The soil map was created by running the multinomial logistic regression (MNLR; [31]) algorithm on the training dataset in R. Map evaluation was conducted with the evaluation dataset, for both the entire Halfway House Granites and the Cosmo City site. An observation was deemed to be correctly mapped if a match was found within the target pixel, or the eight surrounding it [32]. Total evaluation point accuracy, user's and producer's accuracy, and the Kappa coefficient were determined for the evaluation dataset to measure whether the map was an acceptable representation of reality. Total evaluation point accuracy is the total number of observations correctly mapped, expressed as a percentage of the total number of evaluation observations. The user's accuracy reflects the accuracy of the map from the user's perspective. It is the number of evaluation observations correctly mapped within a specific map unit, expressed as a percentage of the total number of observations found on that specific map unit. The producer's accuracy reflects the accuracy of a map from the producer's perspective. It is the number of evaluation observations within a specific class that were correctly mapped, expressed as a percentage of the total number of observations within that specific class. The Kappa coefficient represents how well the map reflects reality when compared to a random designation of mapping units. Kappa coefficient values range between 0 and 1, with values close to 0 indicating that the map is equal to a random designation and values close to 1 indicating that the map represents reality significantly better than a random designation would.

### 2.2.3. Hydrological Modelling

**Kampala Crescent**

The lateral fluxes of Kampala Crescent were quantified in Hydrus 2-D v 2.05 [33]. The location of the hillslope together with the conceptual hydrological response model created from the DSM soil map is presented in Figure 4. The material distribution reflects the distribution of the different soil horizons

according to the South African soil classification system [24], as presented in the conceptual model. The soil hydraulic properties, measured in the laboratory or in situ with the Guelph permeameter for the different soil horizons, are presented in Table 2 together with the Van Genuchten parameters estimated from the hydraulic properties using Rosetta [34]. For the simulation, the hillslope was first saturated through a zero-pressure head at the surface for 10 days and then reduced to −400 mm for 30 days until seepage from the seepage face at Kampala Crescent ceased. A rain event was simulated by applying 1 day of zero pressure on the surface. Thereafter, the boundary condition at the surface was changed to an "Atmospheric Boundary", with 6 mm.day$^{-1}$ of potential evaporation from the surface.

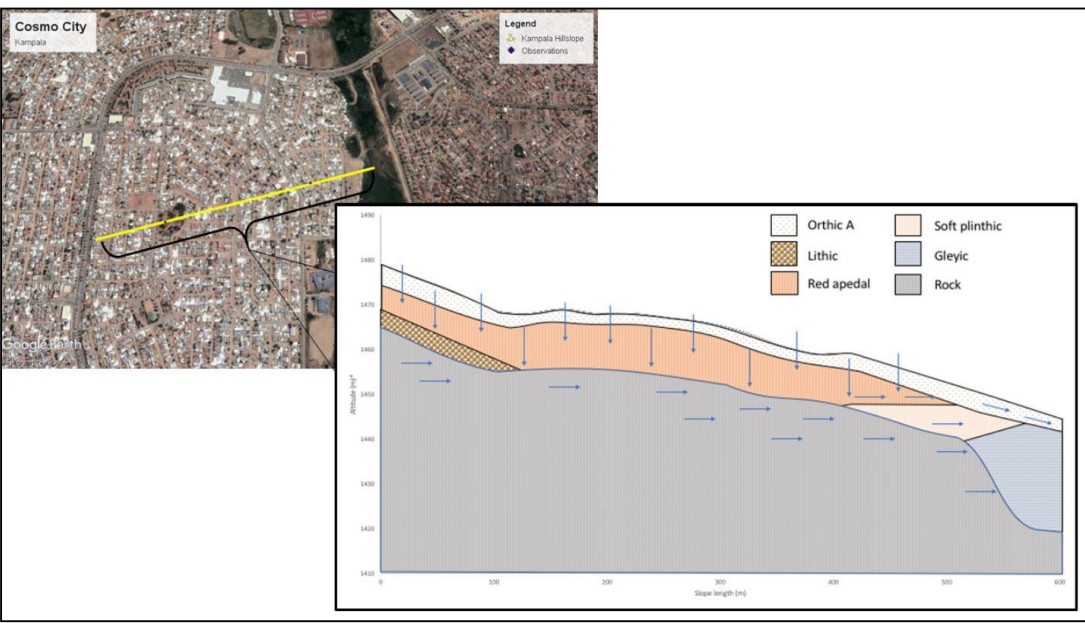

**Figure 4.** The conceptual hydrological response model for the Kampala Crescent hillslope.

**Cosmo City**

The effect of urbanization on the hydrology of the Cosmo City area was simulated using the Soil & Water Assessment Tool (SWAT). The SWAT is a widely used small watershed-to-river-basin-scale model. It is typically used to simulate the quality and quantity of surface and ground water and to predict the environmental impact of land use, land management practices, and climate change. Here we used QSWAT+ (version 0.9).

The catchment area was determined from a 30 m DEM and subdivided into nine sub-basins, each with their own monitoring points (Figure 5). The current land use was obtained from the 2013–2014 SA National Land-Cover Map dataset [35]. The land-cover was re-grouped into SWAT land-uses with pre-defined parameters for each use (Figure 6a). For the historic (before urbanization), we reclassified the urban areas to "grassland" and used the associated parameters. Soil information was obtained from the hydropedological survey (Figure 6b). Hydraulic parameters were derived from in situ and laboratory measurements (Table 3).

A nine-year simulation period was selected (1 August 2003–31 December 2012). Climatic data for this period was obtained from the Climate Forecast System Reanalysis [36] project done by the National Center for Environmental Prediction (NCEP). WeatherGen in SWAT+ Editor used daily precipitation, temperature (minimum and maximum), wind speed, solar radiation, and relative humidity from sta2607s2781e (lat. −25.76, lon. 27.50) to generate daily climatic variables for the simulations. Figure 7 presents the average monthly minimum and maximum temperatures as well as the total monthly rainfall for the simulation period. Results are reported from January 2004, which allowed the model 5 months to settle.

**Table 2.** Hydraulic properties and Van Genuchten parameters of dominant soil horizons used in Hydrus simulations.

| Horizon | Hydraulic Properties | | | | | | | | Van Genuchten Parameters | | | | | |
|---|---|---|---|---|---|---|---|---|---|---|---|---|---|---|
| | Db | FC | DUL | Θs | Ks | Sand | Silt | Clay | Θr | Θs | Alpha | n | Ks | lambda |
| | g·cm$^{-3}$ | mm·mm$^{-1}$ | mm·mm$^{-1}$ | mm·mm$^{-1}$ | mm·h$^{-1}$ | % | % | % | | | | | | |
| Orthic A | 1.39 | 0.2 | 0.22 | 0.477 | 237.2 | 67.6 | 11 | 22 | 0.1 | 0.44 | 0.00315 | 1.4802 | 237 | 0.5 |
| Red Apedal | 1.42 | 0.22 | 0.24 | 0.464 | 73.9 | 56.6 | 15 | 29 | 0.1 | 0.43 | 0.00294 | 1.4273 | 73.9 | 0.5 |
| Soft Plinthic | 1.55 | 0.27 | 0.35 | 0.416 | 2 | 39.7 | 14 | 46 | 0.1 | 0.42 | 0.00216 | 1.2054 | 4.1 | 0.5 |
| Gleyic | 1.55 | 0.27 | 0.35 | 0.416 | 1 | 27.6 | 20 | 53 | 0.1 | 0.41 | 0.00231 | 1.2042 | 1.2 | 0.5 |
| Lithic | 1.26 | 0.09 | 0.28 | 0.526 | 65.7 | 55.4 | 14 | 31 | 0.1 | 0.49 | 0.00271 | 1.3588 | 65.7 | 0.5 |
| Saprolite/Bed rock | | | | | | | | | 0.1 | 0.26 | 0.00258 | 1.1497 | 0.1 | 0.5 |

Db: Bulk density; Θr: Residual water content; FC: Field Capacity; Θs: Saturated water content; DUL: Drained Upper Limit; Alpha: inverse of air entry; Θs: Saturated water content; n: pore size distribution; Ks: Saturated hydrologic conductivity; Ks: Saturated hydrologic conductivity; lambda: Constant.

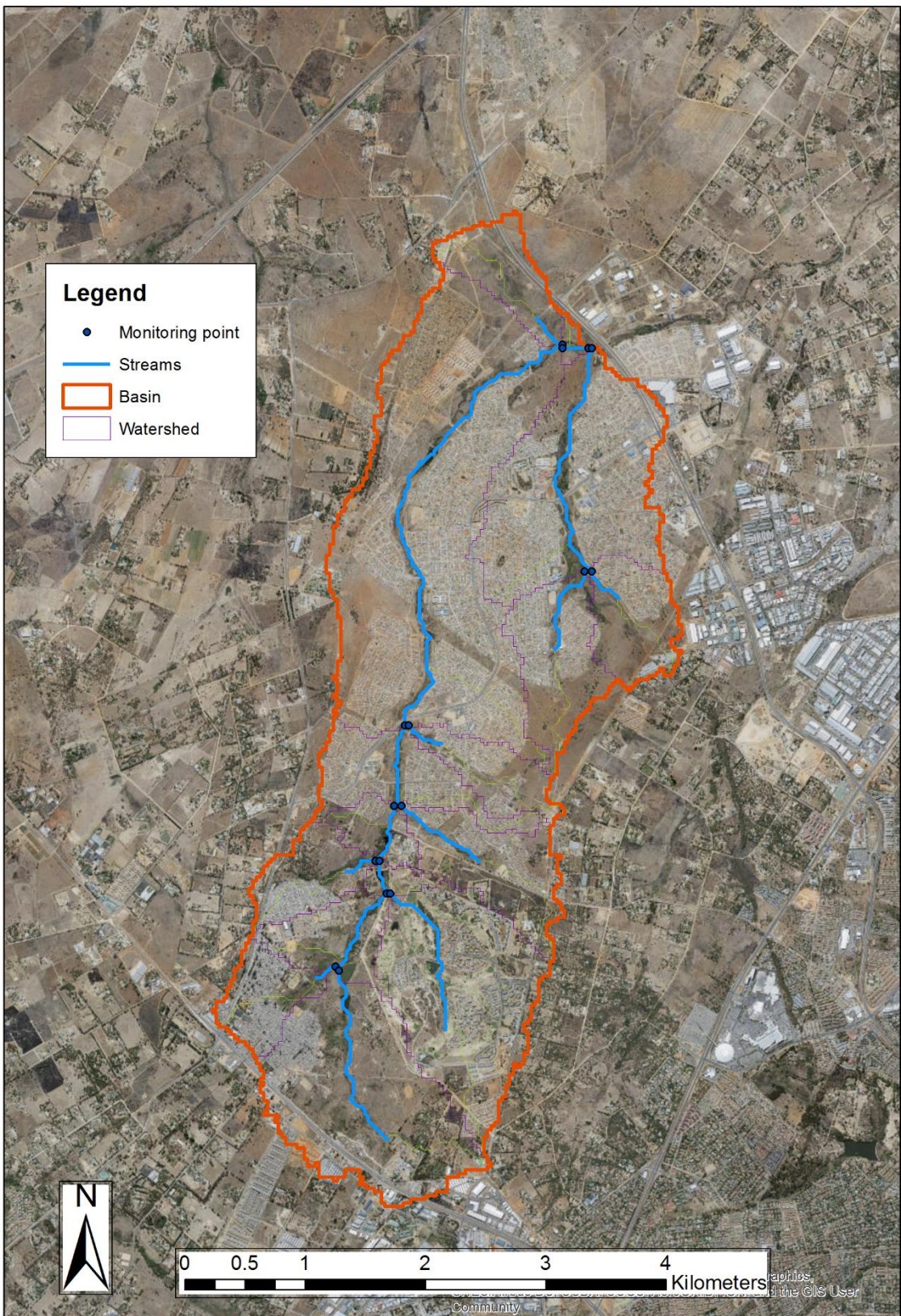

**Figure 5.** Cosmo City catchment, monitoring points, and sub-basins.

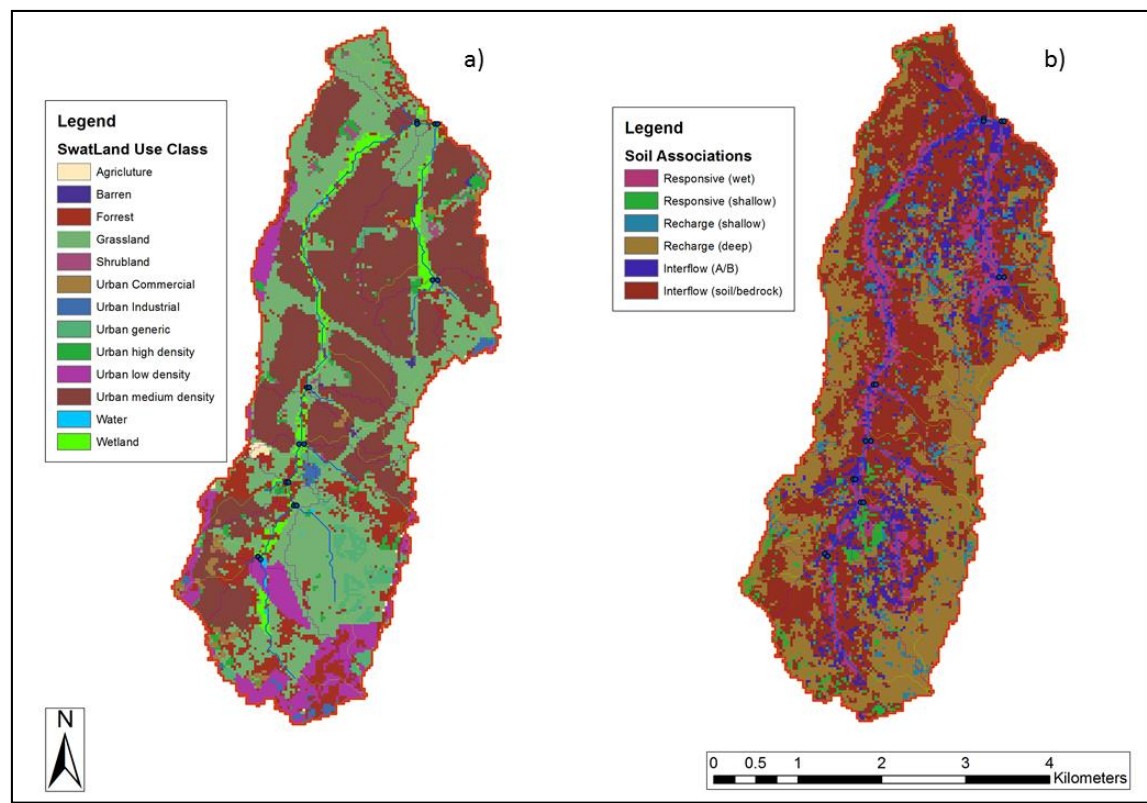

**Figure 6.** (**a**) Swat land use classes, converted from 2013/14 land-cover dataset [35] and (**b**) soil associations obtained from the hydropedological survey.

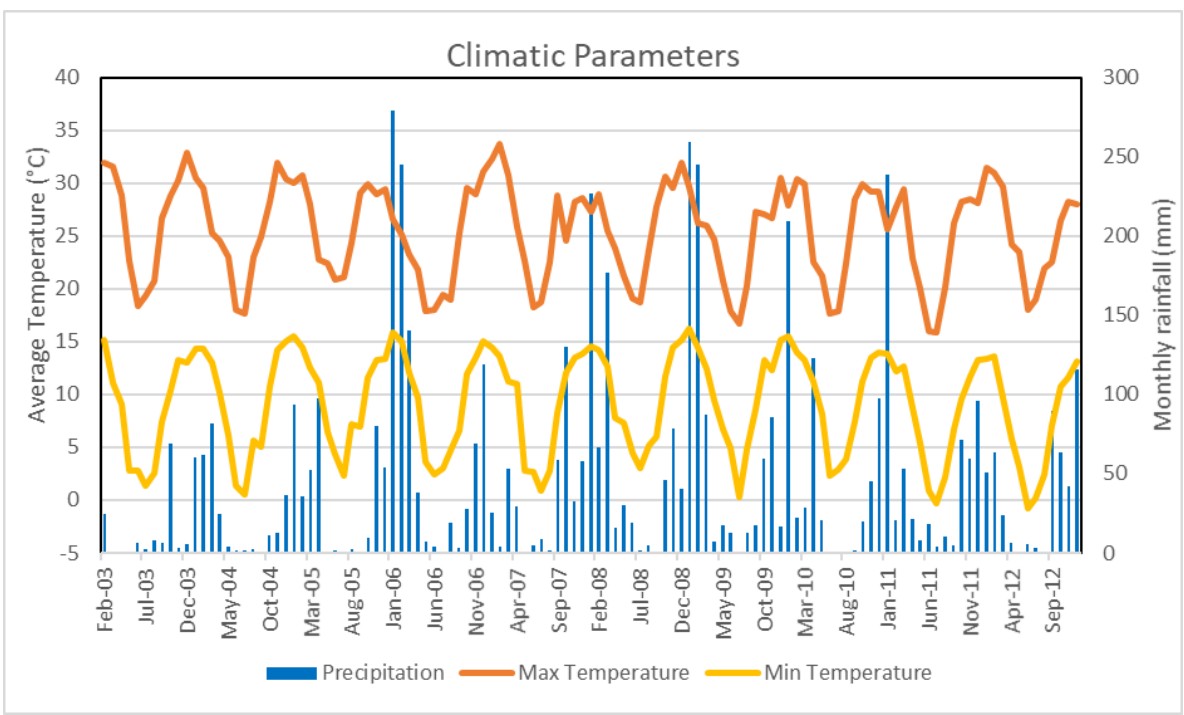

**Figure 7.** Key climatic parameters used in the modelling (average monthly minimum and maximum temperatures and total monthly rainfall).

**Table 3.** Selected hydraulic properties for the soil associations in Figure 6b.

| Soil Association | Hydrological Groups | Layer | Depth | Bulk Density | Wilting Point | Ks | Carbon | Clay | Silt | Sand |
|---|---|---|---|---|---|---|---|---|---|---|
| | | | mm | g·cm$^{-3}$ | mm·mm$^{-1}$ | mm·h$^{-1}$ | | | % | |
| Interflow (A/B) | C | 1 | 300 | 1.4 | 0.06 | 237 | 1.87 | 21.6 | 11.1 | 67.6 |
| | | 2 | 600 | 1.4 | 0.09 | 73.9 | 0.6 | 29.1 | 14.7 | 56.6 |
| | | 3 | 1200 | 1.5 | 0.08 | 2 | 0.5 | 46.2 | 14.2 | 39.7 |
| Interflow (Soil/bedrock) | C | 1 | 300 | 1.4 | 0.06 | 237 | 1.87 | 21.6 | 11.1 | 67.6 |
| | | 2 | 1200 | 1.4 | 0.09 | 73.9 | 0.6 | 29.1 | 14.7 | 56.6 |
| | | 3 | 1500 | 1.5 | 0.08 | 2 | 0.5 | 46.2 | 14.2 | 39.7 |
| Recharge (deep) | A | 1 | 300 | 1.4 | 0.06 | 237 | 1.87 | 21.6 | 11.1 | 67.6 |
| | | 2 | 1200 | 1.6 | 0.07 | 284 | 0.6 | 29.7 | 13.2 | 57.2 |
| | | 3 | 1500 | 1.5 | 0.09 | 73.9 | 0.6 | 29.1 | 14.7 | 56.6 |
| Recharge (shallow) | A | 1 | 300 | 1.4 | 0.06 | 237 | 1.87 | 21.6 | 11.1 | 67.6 |
| Responsive (wet) | D | 1 | 300 | 1.4 | 0.06 | 237 | 1.87 | 21.6 | 11.1 | 67.6 |
| | | 2 | 1000 | 1.5 | 0.09 | 1 | 1.87 | 52.8 | 19.6 | 27.6 |
| Responsive (shallow) | C | 1 | 300 | 1.4 | 0.06 | 237 | 1.87 | 21.6 | 11.1 | 67.6 |

## 3. Results and Discussion

### 3.1. Digital Soil Mapping

The Halfway House Granites hydropedological soil map (Figure 8) achieved an evaluation point accuracy of 80% and a Kappa statistic value of 0.71, indicating a substantial agreement with reality. These results are higher than expected, as lower accuracy was expected due to the urban environment mapped. The map should only be used in areas with a high observation density, such as Cosmo City.

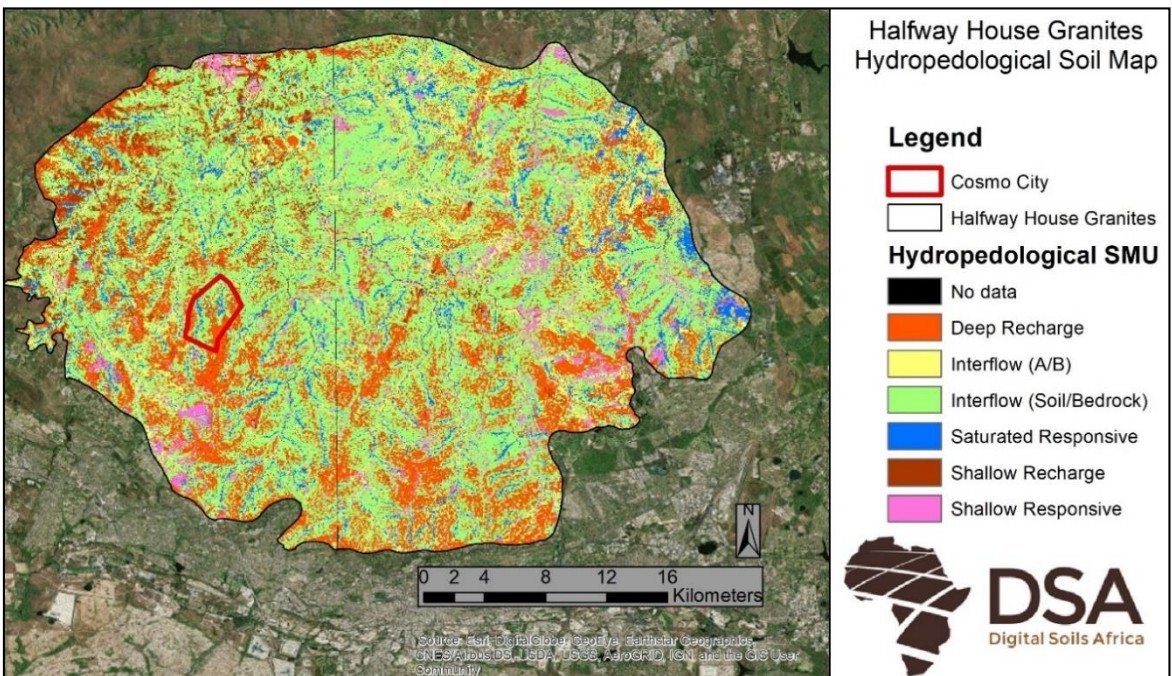

**Figure 8.** The hydropedological soil forms for the Halfway House Granites area. (SMU: Soil Mapping Unit)

The soil map for Cosmo City (Figure 9) achieved even better results (Table 4), with an evaluation point accuracy of 88% and a Kappa statistic value of 0.82, indicating an almost perfect agreement with reality [37]. The user's and producer's accuracies of the map units were all high as well, except for the producer's accuracy of the shallow recharge soil association, which was only 33%. This was due to there being only three shallow recharge observations. Overall the results for Cosmo City are very satisfactory, and compare very well with maps created with similar methods, such as Van Zijl et al. (2019) [28], 69% point accuracy, Kappa = 0.59; Van Zijl et al. (2012) [38], 69% point accuracy, Kappa = 0.55; MacMillan et al. (2010) [39], 69% point accuracy; Zhu et al. (2008) [40], 76% point accuracy. The Cosmo City map achieved better results, as the observation density was higher than the rest of the Halfway House Granites area, and the satellite images showed the natural state of the area before development commenced. However, due to the relatively low number of evaluation observations the accuracy assessment should only be taken as indicative.

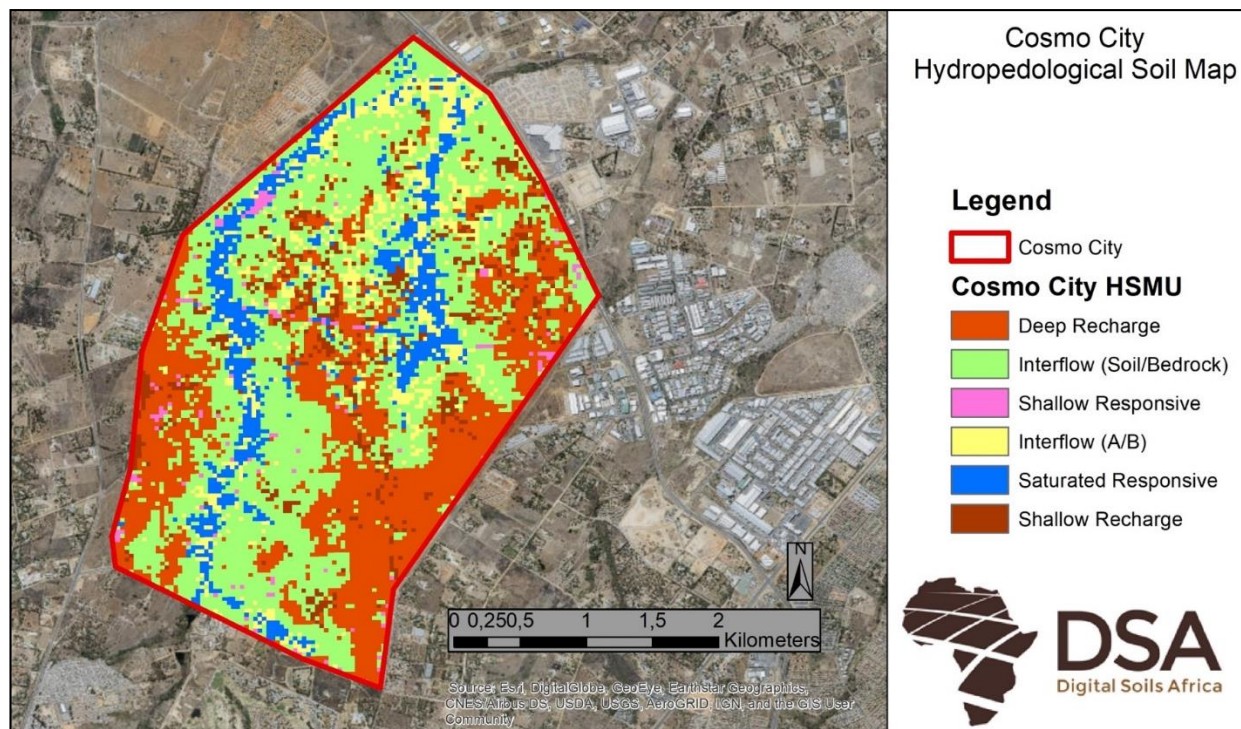

**Figure 9.** The Cosmo City hydropedological soil map (HSMU: Hydropedological Soil Mapping Unit).

**Table 4.** Confusion matrix for the Cosmo City hydropedological soil map.

| | | Deep Recharge | Shallow Recharge | Users Accuracy Interflow (Soil/Bedrock) | Interflow (A/B) | Saturated Responsive | Total | Correct | % Accuracy |
|---|---|---|---|---|---|---|---|---|---|
| **Producer's Accuracy** | **Deep Recharge** | 3 | | | | | 3 | 3 | 100 |
| | **Shallow Recharge** | 1 | 1 | 1 | | | 3 | 1 | 33 |
| | **Interflow (Soil/Bedrock)** | | | 8 | | | 8 | 8 | 100 |
| | **Interflow (A/B)** | | | | 1 | | 1 | 1 | 100 |
| | **Saturated Responsive** | | | | | 2 | 2 | 2 | 100 |
| | **Total** | 4 | 1 | 9 | 1 | 2 | 17 | | |
| | **Correct** | 3 | 1 | 8 | 1 | 2 | | 15 | |
| | **% Accuracy** | 75 | 100 | 89 | 100 | 100 | | | 88 |

### 3.2. Modelling Results

#### 3.2.1. Kampala Crescent

The 2-D simulations of Kampala crescent demonstrate that seepage will occur immediately after the rain event from the 9.3 m seepage face (Figure 10). The maximum seepage rate is approximately 32 mm·h$^{-1}$ and occurs after 6 h of rain. The seepage rate will decline as a 6 mm·day$^{-1}$ potential evaporation until seepage will cease after approximately 8 days. With a peak discharge rate of 32 mm·h$^{-1}$ the total volume of seepage water from in the area is approximately 0.3 m$^3$·h$^{-1}$·m$^{-1}$ of roadway (with the 9.3 m seepage face). If, for example, 100 m of Kampala Crescent is impacted by the seepage, subsurface drains should have a carrying capacity of 30 m$^3$·h$^{-1}$ (0.085 L·s$^{-1}$·m$^{-1}$).

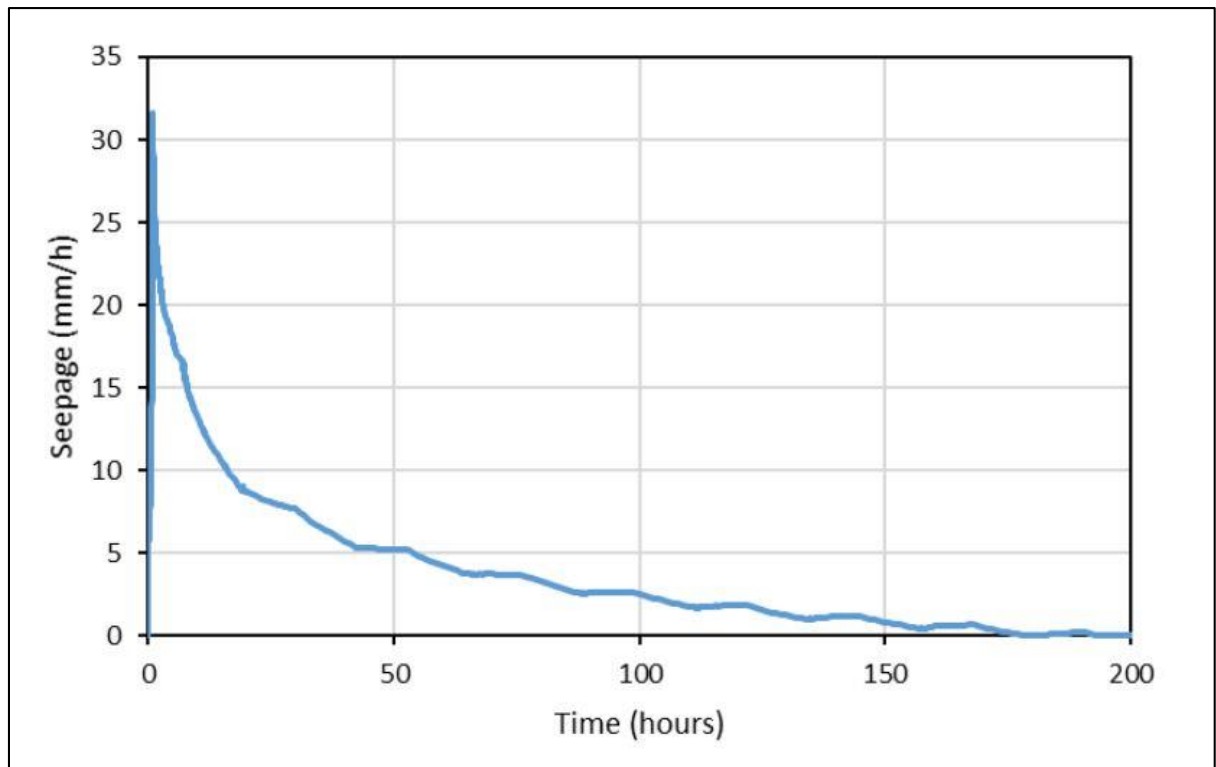

**Figure 10.** Resultant seepage hydrograph with one hour of rain at peak infiltration rate.

#### 3.2.2. Cosmo City

The effect of urbanization on the hydrology is clearly shown in Figure 11, which shows the modelling results for the period from November 2011 to November 2012. The total streamflow from the Cosmo catchment increased due to the surface runoff also increasing. This effect is due to the surface sealing occurring with urbanization. Additionally, the deep percolation and lateral flow decreased as less water infiltrated the soil. Specifically important is the observation that, under natural conditions, surface flow was only initiated when monthly rainfall exceeded 50 mm·month$^{-1}$. However, after urbanization this figure dropped to 2 mm·month$^{-1}$. This impacts storm water management, as higher flow rates will occur, and they will also occur more frequently.

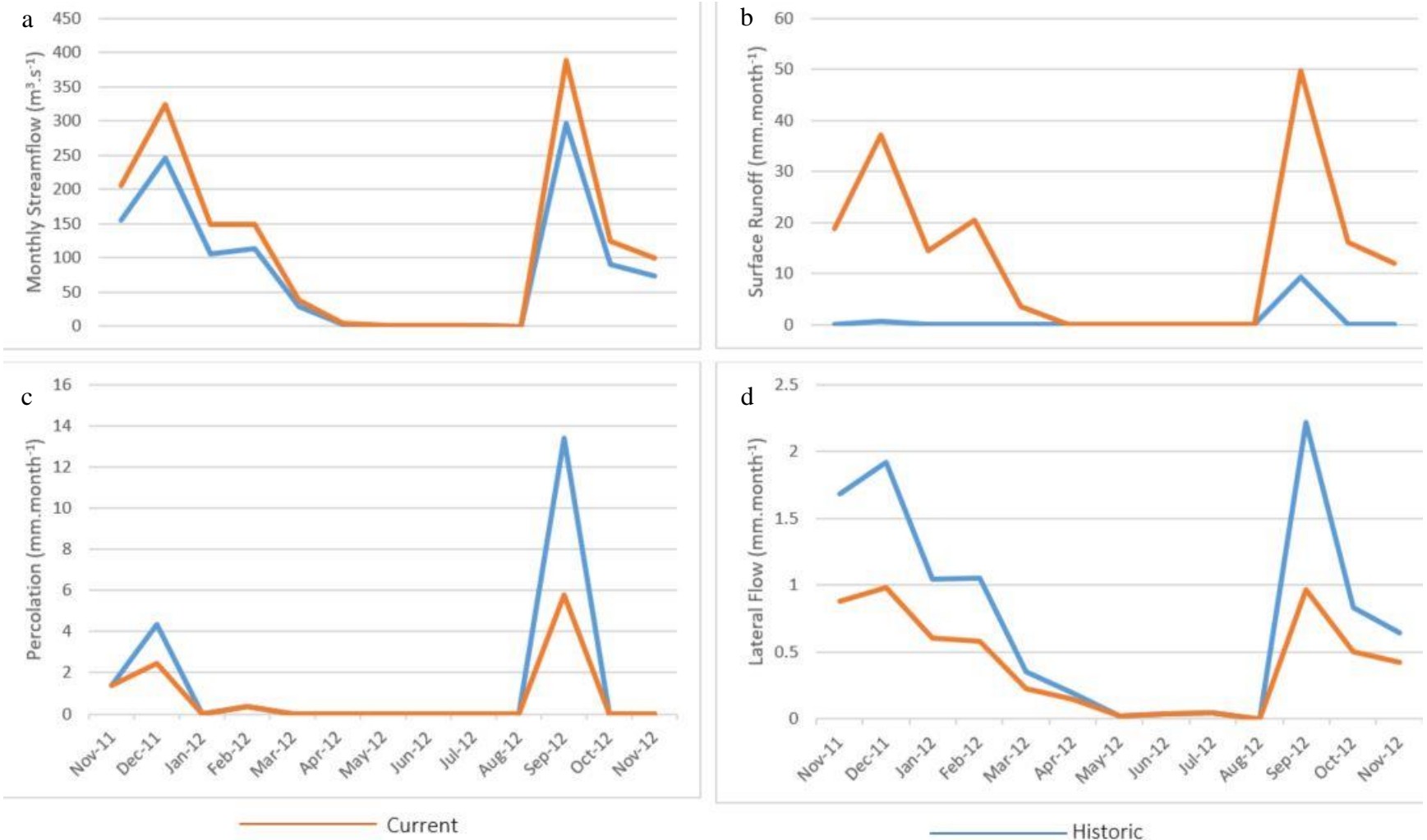

**Figure 11.** Average monthly modelling results for the Cosmo City suburb. (**a**) Streamflow, (**b**) Surface runoff, (**c**) Deep percolation, and (**d**) Lateral flow, for the time period November 2011 to November 2012.

## 4. Conclusions

Using historical satellite images from before the Cosmo City suburb was developed, and combining these with DSM methods, a natural-conditions hydropedological soil map could be created for the suburb, with a Kappa statistic value of 0.81. This map could then be used to parameterize both the HYDRUS and SWAT hydrological models. HYDRUS modelling was used to determine the source and magnitude of soil water causing structural damage at the Kampala Crescent, and it was determined that a subsurface drain with a carrying capacity of 30 $m^3 \cdot h^{-1}$ (0.085 $L.s^{-1} \cdot m^{-1}$) should solve the problem. SWAT modelling revealed the effects of the development on the hydrology of the area. The development and subsequent surface sealing of the area caused an increase in runoff and streamflow, but reduced the evapotranspiration, lateral flow, and deep percolation. The hydrological impacts should be considered when designing new urban developments.

This case study demonstrates the value of historic remote sensing data, as well as the power of multidisciplinary work. The remote sensing was invaluable to the mapping, which informed the hydrological modelling, which provided answers to the engineers, who could then mitigate the hydrology-related issues within Cosmo City.

**Author Contributions:** The authors made the following contributions to the work: Conceptualization, G.v.Z., D.B., S.L., J.v.T., and P.l.R.; Methodology, G.v.Z., D.B., S.L., J.v.T., and P.l.R., Data Acquisition, G.v.Z. and D.B., Formal Analysis, G.v.Z., D.B., S.L. and J.v.L.; Writing—Original Draft Preparation, G.v.Z.; Writing—Review and Editing, J.v.T.; Project Administration, G.v.Z. and D.B., Funding Acquisition, G.v.Z., D.B., P.l.R., and J.v.L. All authors have read and agreed to the published version of the manuscript.

**Funding:** This study was funded by the Johannesburg Roads Agency.

**Acknowledgments:** We would like to acknowledge the Johannesburg Roads Agency and Digital Soils Africa who allowed this data to be used for this publication.

**Conflicts of Interest:** The authors declare no conflicts of interest, and the funder played no role in the choice of research project; design of the study; in the collection, analyses, or interpretation of data; in the writing of the manuscript; or in the decision to publish the results.

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
