# Peer review of "Combining Historical Remote Sensing, Digital Soil Mapping and Hydrological Modelling to Produce Solutions for Infrastructure Damage in Cosmo City, South Africa"

_remotesensing, doi:10.3390/rs12030433_

Round 1
Reviewer 1 Report
Summary
The HYDRUS-2D and SWAT hydrological models were to be used in evaluating urban hydrology condition in Cosmo City, RSA, but the underlying natural soil (i.e., where pavement and housing had not covered it) was not mapped. Nor was anything known about throughflow below sealed areas. So a retrospective DSM of soil functional hydrology types was created with the help of historical imagery.
Evaluation:
This paper is a nice example of practical multidisciplinary work to solve a real social/environmental problem. As such it can be an inspiration and guideline for similar work elsewhere. Perhaps it is more suited to an urban hydrology or urban planning journal, because remote sensing doesn't play a large role -- although the role it does play is vital. The digital soil mapping is "retrospective" -- i.e., the situation before urbanization, which can't be observed now, using imagery before land conversion and observations only in supposed undisturbed areas. This is clever and a key part of the hydrologic modelling. The categotization of hydrological functional types is also clever.
Recommendation:
Minor revision, see points below.
Specific points:
Title: remote sensing is not mentioned, yet this is an RS journal. Maybe too long a title, but could you add "historical remote sensing and DSM to..."?
L23 "in combination with" terrain parameters and "digital soil mapping"....
L26 "a road within Cosmo City (HYDRUS 2D)" -- what is HYDRUS 2D doing here? Also, the explanation ""a road within Cosmo City" should be added at the first mention in the main text (L153). And it is not clear why this road is chosen. Fig. 1 the road is just a dot. Fig. 4 shows a yellow line across the hillslope but which is the road? Is it the large curved road above the yellow line? In short, I can not see where is Kampala Crescent and where the drain might be.
L28, 259: why report this without the dimensions? How can you assume 100 m of roadway (L238)? Is it not better to report the carrying capacity per meter?
L61-2 "interpolate between soil observations" not really, it allows unobserved locations whether between or outside observations to be predicted, based on observed points and their relation with covariates.
L63 "This paper presents how remote sensing was used to provide the tools required to use DSM" well, RS was only a part of the DSM, you also used the conventional terrain covariates, which obviously are important in grouping hillslope soils. The point is that historical RS gave you the pre-urban land cover. It's not clear how the Landsat 8 (current) helped.
L75 outside of RSA the technical term "soil form" will not be clear
L76 " the vegetation of Cosmo City has been cleared to" was there also land shaping? compaction?
L97-8 more details of methods. Maybe just to a standard methods book used in RSA?
L110 Landsat 8 -- but these are contemporary (after construction) -- what information are they expected to provide?
L113 more details of methods, maybe there is another paper that used the same covariates as computed in SAGA? Needs a reference to SAGA.
L115 for mapping purposes --> no, this is for modelling purposes. You then use the model to map within Cosmo City.
L135 MNLR OK but why this choice? A random classification forest could have been used on all the points, it would give OOB evaluation, no need to split train/test (see comment below on Table 4).
L137 "A one pixel buffer was observed" I think you mean that the evaluation was considered correct if a match was found within the 9 pixels including the supposed target?
L139, L147ff. Kappa has nothing to do with reality, it measures the mapper's skill vs. a random allocation of classes (as explained in part of L147-8). The accuracy measures are what affects the map user, this is the agreement with reality.
L161 Rosetta --> needs a reference: Schaap, M. G., Leij, F. J., & van Genuchten, M. T. (2001). ROSETTA: a computer program for estimating soil hydraulic parameters with hierarchical pedotransfer functions. Journal of Hydrology, 251(3–4), 163–176. https://doi.org/10.1016/S0022-1694(01)00466-8
L163 "9.3 m seepage face" where does this come from? Is this the road margin? Is it an upslope drain to protect the road?
L183 Table.... ? (3 I think)
L209 the evaluation is with respect to the original sampling plan, and this statement must be qualified if referring to success over the study area.
L210 "These results are higher than expected" why? It's good but others have achieved similar evaluation results with good covariates. Maybe this refers forwrd to L221 ff.
L217 "almost perfect agreement with reality" ?? to me 88 < 100 and not even approx. equal to
Fig. 9 the lack of connectivity in some classes is worrying... obviously the result of the DSM model... does this seem realistic?
Table 4 I appreciate the effort but with so few evaluation points it's very hard to be confident in the statistics. Just changing one cell would have a fairly large effect. So I would just call this indicative and not emphasize it much.
However, if out-of-bag (OOB) cross-validation had been used with either a random forest, or repeating the MLR with each point kept out in turn, this matrix would have many more observations. With so few observations total, I would strongly recommend this. Although it will likely not change the map much, it would give more reliable evaluation statistics.
Fig. 11 is quite important but not clear, the dotted lines are difficult to see. Maybe one thick and one thin but both solid? I am not a graphics specialist.
\S4 Conclusions: L254-263 repeat the abstract. The real conclusion is L264ff.
References: check carefully, for example ref. (25) "Granites" in different font than the rest of the title, Ref. (27) dutch -> Dutch; I did not check all of them, that is the authors' job
Reviewer 2 Report
Dear Authors,
I think that your manuscript entitled “ Using Digital Soil Mapping to create a natural conditions soil map for hydrological modelling, in an urban setting” is quite systematic and scientifically interesting, as well as brings new and practical applications.
This work is interesting from both scientific and practical point of view, it is also tidy, hence it is suitable for publication in Remote Sensing, after a minor but necessary revision.
1 The title is long and awkward, however what should be necessarily changed also too general. You performed studies on given object - Cosmo City, so add it as study case, please.
Line 69: The dot is missing. Line 114: Was NDVI vegetation index enough for such analyses? You only wrote, "From the wet and dry season satellite images the normalised difference vegetation index (NDVI) were derived." Explain better this part of the analyses. What was the effect of changes in NDVI for the final results? How big changes were observed and with what accuracy? Finally, why NDVI was chosen? Line 137: A one-pixel buffer was observed [28]. Please, describe better in short what it does mean?
4 Lines 141-150: The authors describe accuracy, user's and producer's accuracy and the Kappa coefficient, which are basic terms in remote sensing. I would suggest removing this detailed description, if not give references, please, because this obvious information is not a result of the study. Here, you can reduce the manuscript.
Fig. 11: I would suggest drawing the comparison of current and historic results using different colors. It would be much comfortable for readers. The units on the Y-axis are difficult to see. Make the figure caption with the units.
However, it is even more important to describe better the results shown in Figure 11 in detail, because there are a lot of astonishing details that should be clarified for observant readers.
For example surface current runoff shown in Figure 11b in Jan-06 and Jan-09 are similar, but at the same time, historical runoff is completely different. Why? Besides, in Figure 11b practically only in Jan-09 and slightly in Jan-10 the surface runoff is a non-zero one. What about the other periods? Why does so big peak appear in Jan-06? The similar remarks concern Figures 11 c and b, but they are very unclear because it is difficult to distinguish both lines. Could you mark them better and put the error marks?
Because Figure 11 shows one of the most important results of the study I would suggest to magnify individual parts of this Figure, namely a, b, c, and d, arrange them vertically, and describe each of them them in detail, in the studied period.
The references, especially in the introduction are scarce and quite old. There were various much more attempts to use Landsat images for hydrological analyses on urban, industrial or mining areas. I would suggest you consider supplementing the bibliography especially with the last achievements in the field. Otherwise, readers obtain very limited information on possibility of using Landsat imagery for such or similar analyses. Line 263: Replace Thee by The.
Your sincerely,
Reviewer
